# Green Social Prescribing in Practice: A Case Study of Walsall, UK

**DOI:** 10.3390/ijerph20176708

**Published:** 2023-09-04

**Authors:** Qian Sun, Mary Loveday, Saw Nwe, Nike Morris, Emily Boxall

**Affiliations:** School of Design, Royal College of Art, Kensington Gore, London SW7 2EU, UK; marylovedayedwards@gmail.com (M.L.);

**Keywords:** Green Social Prescribing, Social Prescribing, place-based thinking, case study, link worker, Walsall

## Abstract

This paper presents a case study of Green Social Prescribing (GSP) in Walsall, a medium-sized urban area located in the West Midlands, UK. GSP is a means of enabling health professionals to refer people to a range of local non-clinical nature-based activities, e.g., community gardening and conservation volunteering. As a new practice to address multiple challenges in health and sustainability, GSP has been promoted by the UK government and the NHS in the past few years. There is as yet limited evidence and knowledge about how this approach is implemented at a local level. This paper addresses this gap of knowledge, by exploring how GSP is implemented in Walsall as a case study. Based on extensive engagement and research activities with the local partners to collect data, this paper reveals the local contexts of GSP, the referral pathways, and people’s lived experience, discussing the challenges, barriers, and opportunities in delivering GSP at the local level. This study suggests that a more collaborative and genuine place-based approach is essential, and alongside GSP, investment into infrastructure is needed to move the health paradigm further from ‘prevention’ to ‘promotion’ so that more people can benefit from what nature can offer.

## 1. Introduction

Experience with the natural world benefits physical and mental health. It is associated with a sense of gratitude and self-worth [1] and can help people recover from stress and mental illness [2]. This kind of experience with nature also helps to build a sense of place and community and foster feelings of belonging [3]. There is ample evidence of well-being pathways from nature [4,5,6,7,8,9]. GPs, nurses, and other healthcare professionals can prescribe nature-based activities as a part of NHS services at no direct cost to users, such as local walking for health schemes and community gardening to those who could benefit from them. This is called Green Social Prescribing (GSP) [10], part of the Social Prescribing practice sometimes known as community referral, a means of enabling health professionals to refer people to a range of local non-clinical services. These nature-based activities include a diverse range of activities, e.g., green exercise (such as local Park Runs and dementia walks), active travel (such as cycling), care farming, community gardening and food growing projects, conservation volunteering, and Green Gyms.

The COVID-19 pandemic not only increased people’s awareness of the importance of nearby nature, but also exacerbated long-standing problems of health inequalities [11]. With the imperative of the NHS to improve health outcomes with increasingly limited resources, the asset-based approach of GSP, in drawing on locally available resources to deliver healthcare services to local communities, is seen as vital for the future viability of the NHS in the UK. The NHS five year forward view [12] sets out a vision of how NHS services need to change to meet the future needs of the population, arguing for greater emphasis on prevention, integration, and putting patients and communities in control of their health. In the policy vision, the emphasis is on health promotion [13] and prevention, with the aim of enabling people to increase control over and to improve their health, and to prevent the initial occurrence of a disorder, e.g., through behaviour change advice and prompting. This paper aims to describe findings from a project with a place-based research approach, arguing that local detail provides the necessary context for this policy vision.

The activities in green social prescriptions are typically free. They are either provided by nature itself—it is free to spend time in public nature spaces, for example—or by community or voluntary groups. For more involved activities, such as a group trip to Snowdon provided by one of the organisations in the study, for example, a small charge is sometimes made.

Referring people to nature-based activities has been practised in pockets for some years in the UK, although the NHS has only recently started to embrace this and has committed resources to rolling out GSP schemes across England. Public Health England, in their report ‘Improving Access to Greenspace: A new review for 2020’ [14], recognises local green (and blue) space as critical assets for maintaining and supporting health and well-being in local communities. In July 2020, NHS England announced a GBP 4 million investment for a cross-government project aimed to “improve mental health outcomes, reduce health inequalities, reduce demand on the health and social care system, and develop best practice in making green social activities more resilient and accessible” [15].

With the scale of rolling out GSP across England for the past few years, there is emerging knowledge from these early pilots and projects at local levels. It is timely to review and to capture the learnings to provide formative feedback for the application of GSP both locally and nationally. We ask: what can we learn from the current implementation and adaptation of GSP at local levels? This paper aims to develop a case study to unpack these initial learnings to understand how green Social Prescribing is experienced at the local level in order to identify the challenges and opportunities for GSP. With a case study approach, this paper provides in-depth knowledge that complements the growing body of research into the benefit of and need for GSP, which is often approached through exploring the role of nature in well-being in systematic reviews or meta-analyses [16].

### 1.1. The State of the Art

#### 1.1.1. Green Social Prescribing

GSP is an interdisciplinary concept, suggesting a healthcare system that integrates and maximises the value of what society and nature can offer to public health. It is rooted in two distinct areas of public health research: (1) one explores the integration of community-based services into the healthcare system through Social Prescribing, and (2) the other explores the value and implementation of nature-based activities as a pathway for prevention.

#### 1.1.2. Community-Based Services

Early studies [17,18] suggested that many problems brought by patients to GPs could have potentially been dealt with more effectively by the wider primary care team, such as clinical pharmacists, practice nurses, or physician assistants, or by patients being supported to meet their own health needs [19]. This started the shift of thinking to integrate social care into the delivery of healthcare [20]. There is a wealth of evidence suggesting that social support is therapeutic to the recipient both from the physiological and psychological perspectives [21,22,23,24]. It is recognised that the third sector often fills a gap in services provided by the statutory sector, where experiences, skill sharing, and social networks can provide respite and solace for those not satisfied by mainstream channels [25]. Community assets—the positive capabilities within communities—are therefore seen as an important element in public health [26]. In England, this vision is emphasised in the NHS strategic plan [27] that sees partnerships with the third sector as a productive means to create more value from healthcare services and employ resources effectively. The recognition represents a major shift in public health thinking for both the prevention and treatment of health issues [28].

This has led to the rolling out of Social Prescribing practices in England—a process of linking individuals to social or community-based activities or resources that have the potential to improve health and well-being [29]. The establishment of a link worker (LW) role—someone who serves as the core contact person for someone who has been prescribed—has been integral to Social Prescribing. Linking pathways expand the options available to individuals who have complex social as well as clinical needs, by connecting people to community resources, information, and social activities. The Marmot report [30] clearly recognised Social Prescribing as part of a policy initiative to engage with health inequalities. From the perspective of local councils, it is a way of reducing visits to the doctor and hospital, and of ensuring people receive the most appropriate support they need to take control of maintaining or improving their own health and well-being [31].

#### 1.1.3. Nature-Based Services

Green Social Prescribing (GSP), a form of Social Prescribing, builds on a longer-term tradition of therapeutic interventions, such as “green care” [32], “horticultural therapy” [33] and “nature assisted therapy” [34,35], broadly termed as “nature-based health interventions” [29] or “nature-based health services” [11]. There is ample epidemiological evidence demonstrating that experiencing nature is associated with multiple dimensions related to human health: physical, psychological-emotional, social, and spiritual [7,36,37,38,39,40,41,42,43]. The positive association applies in general as well as specifically to older adults [44], children, and young people [45]. Irvine and Warber’s (2002) early review and synthesis of the literature [4] concerning the health benefits of nature suggested that as interaction with the natural world is a vital part of biopsychosocial–spiritual well-being, incorporating the natural world into healthcare might change the way we approach public health.

Within the specific context of COVID-19 lockdowns, local green spaces took on a new significance in many people’s lives [46]. RSPB’s survey [47] conducted by YouGov in May 2020 reveals that the COVID-19 pandemic has not only increased people’s awareness of the importance of nearby nature, but has also exacerbated long-standing problems of health inequalities in access to nature and its benefits between households with the highest and lowest incomes, and between urban and rural households. Public Health England, in the report “Improving Access to Greenspace: A new review for 2020”, [14] recognises local green (and blue) space as critical assets—“natural capital” for maintaining and supporting health and well-being in local communities.

As such, there has been a surge of research interest in GSP. Different types of nature-based interventions have been examined, including community gardening [48,49], blue-care or water-based activities [50,51], and general outdoor activities [52]. Some studies in particular explore GSP in the context of COVID-19 [53,54]. These studies have identified a range of challenges and opportunities for research and practice.

### 1.2. Challenges

Despite promising moves towards scaling GSP, early pilots have suggested that delivering GSP has proven challenging. Despite a general positivity towards the concept [55], some GPs and link workers are reluctant to prescribe nature-based health services to their patients or clients due to a range of perceived challenges [29,53,56].

One of these is the lack of convincing evidence that the benefit from nature and community assets can be realised through GSP [57,58]. Another important consideration is the lack of consistent or standardised referral mechanisms, as schemes are funded, structured, and delivered diversely [53], and so the referral process is highly individual [54].

Furthermore, GSP operates within a complex Social Prescribing system made up of arrays of interconnected and interdependent actors, processes, and events [11,29,53,54,59,60]. The complexity and the lack of communication between different parts of the system create significant barriers for individuals and organisations to navigate through the system [50].

While link workers are of fundamental importance in bridging the gap between the primary care professionals and communities [58], they experience several challenges to client engagement, which include training and networking needs; and volume, suitability, and targets for referrals [61]. Meanwhile, across the entire system, funding and capacity are important constraints shared by majority stakeholders, including GPs, link workers, and VCSEs [29,51,57,58,62,63,64]. This is especially the case for VCSEs, who are reliant on human resources—usually volunteers—to meet needs but face the challenges of a lack of recognition, capacity, and necessary investment [11,54,60]. Juster-Horsfield and Bell [51] question whether VCSE sectors in the UK may not be ready to accommodate growing demands for GSP, given no clear or sustainable pathway for meeting key funding, training, or staffing needs.

From the VCSEs’ perspective, the most frequently expressed constraint is the inability to engage with GPs and other primary care professionals, whilst from the GPs and link workers perspective, it is the lack of available options [57].

Fundamentally, GSP relies on the availability of local natural resources, e.g., lakes, parks, etc. [53], as well as on local infrastructure, including the layout of communities, transportation, and access to parks and trails. Barriers include mobility issues and policies around safety risks, but the literature also suggests a range of psychological barriers, including anxieties, motivation, and scepticism [29,56]. Social disadvantage, chronic ill health, and health crises all limit easy access to green and blue spaces [53].

The accessibility—and quality—of green space is beset by significant and continuing issues [65,66,67,68,69,70], which are greater in areas of deprivation [71]. In addition, complications and restrictions relating to green sites, or identified potential green sites, can affect the development and management of nature-based projects [72,73,74], and therefore, local authorities’ role in allocating land use and access to green spaces for local organisations is essential. Local authorities also have a place-based understanding of the green spaces in question.

Both nature-based and community interventions are rooted in place-based thinking, where the concept of “place” is defined as a geographical area that is meaningful to communities who live there on an economic, physical/environmental, or social level. In contrast to discrete interventions addressed at a single particular issue, place-based interventions describe a human-centred and bottom-up, systems-based approach that works with multiple partners to address causes rather than problems, including the wider determinants of health [75,76]. However, it takes time to understand places, to build relationships, and for interventions to be implemented and established [77]. In addition to the long-term nature of projects, the systemic nature of the place-based approach, in which the very local is embedded in wider systems, creates difficulty in discrete reporting on change, which is driven by complex, cross-sectoral factors [78]. Challenges in targeting and monitoring change contain the risk of the approach becoming weakly specified and poorly evidenced [79] to the extent that the approach is considered to require “a leap of faith, to some degree” [80] (n.p.).

### 1.3. The Focus of the Paper

This paper aims to develop a case study of Green Social Prescribing (GSP) to capture the initial learnings from the local initiative and to explore how the challenges, barriers, and opportunities identified in the literature are played out at the local level. In particular, this paper aims to understand (1) the local context for GSP; (2) how the local GSP schemes are structured, resourced, and delivered; and (3) what the lived experience is like for those who are participating in the schemes, including the link workers, users of GSP services, and VCSEs who support GSP.

This sheds light on the ongoing exploration of GSP as an emerging approach to address health challenges at the local level and contributes to the discussion concerning whether the potential of GSP can be realised and scaled up, by understanding the challenges both at the system and individual levels.

## 2. Methodology

### 2.1. Case Study Approach

This paper uses a case study approach that generates in-depth insights into phenomena and engagement with data, allowing for the consideration of the holistic nature of the research [81,82,83]. The project used a qualitatively driven multiple-method approach to collect data to compile the case study, where the combination and synthesis of different kinds of methods within projects minimises each method’s limitations or weaknesses [82,84].

We chose a case study approach to capture both the uniqueness and complexity of the study subject. In a relatively new phenomena such as GSP, the relationship between the context of Walsall, the people involved in its implementation, and GSP itself first needs to be explored and described. Questions such as “how is GSP being implemented, received, and extended in Walsall?”, and “what are the challenges on the ground?” can lead to insights about why one approach or intervention is more successful than another, or what gaps exist in implementation [85]. In a place-based approach, multiple perspectives and contexts—such as historical, sociological, and cultural—can interact to influence the relationship between theory and practice and affect how stakeholders variously experience and respond to phenomena. Capturing these contexts and foregrounding the lived experience of stakeholders can help us to understand and explain the pathways and links resulting from efforts to implement policy. Further, a case study approach allows for the use of a number of methods, including interviews, workshops, site visits, and observations, as explained in Section 2.3, to support the iterative exploration of the data from different angles to provide depth of detail.

The team, comprised of researchers and community partners experienced in interviewing and data analysis, worked closely together to sense-check findings and minimise bias in the analysis. One of the design researchers was also a doctor within the NHS, adding detail and depth to the understanding of the project in the clinical and institutional contexts, and another researcher was embedded in the community and therefore able to identify and verify contexts and detail. This contextual knowledge and understanding is imperative in a place-based approach.

### 2.2. Walsall

Walsall, a medium-sized urban area located in the West Midlands, UK, with an estimated population of 286,700 [86] was chosen as the place to develop the case study. Walsall is an ethnically and culturally diverse town with around a third of the population coming from ethnic minorities. People of Indian, Pakistani, and Bangladeshi background form the largest minority ethnic groups, and now account for 1 in 3 of Walsall’s population. It is within the most deprived 10% of districts in England according to the 2019 Index of Multiple Deprivation [87], with pockets of deprivation existing even in the more affluent parts of the borough. Overall health shows an average ten-year lower life expectancy when compared to that in England and Wales [88], and health inequality is high. Even before COVID-19, about 28.1 percent of Walsall’s population had experienced mental health problems, such as anxiety and depression. Further, Walsall is particularly susceptible to outbreaks of COVID-19, sitting within the 20% most vulnerable local authorities in England according to British Red Cross’ COVID-19 vulnerability index [89]. The scale and severity of the COVID-19 pandemic has accentuated pre-existing inequalities and created new and unprecedented demands on public services [90]. Walsall provides an ideal case study for the introduction of GSP for both community well-being and environmental protection and regeneration. As one of the areas receiving funding as part of the UK government’s levelling up agenda to level up towns and cities around the country and to drive sustainable regeneration for long-term economic growth, this case study sheds light on the relevance of GSP in supporting and benefiting from this scheme.

The team has been working with local partners for a number of years and has established relationships across various sectors, which supports close and collaborative partnerships based on mutual support and trust. This has granted the research team open and honest insights from within Walsall.

The project, Connecting Roots: Co-creating a Green Social Prescribing Network in Walsall for Health and Wellbeing, is a research project funded through AHRC (Arts and Humanities Research Council UK)’s “Mobilising community assets to tackle health inequalities” programme, which is aimed at levelling up health and well-being in the UK. The study took place from August 2022 to August 2023. Although all COVID restrictions had been lifted by this point, some of those interviewed were still reflecting on their experiences of lockdown.

### 2.3. Process

This project has delivered desk research (reviewing key literature and policy documents) and field work through a range of data collection methods including:(1)Visits to sites, such as public green spaces, allotments, and community gardens, to develop a holistic understanding of the place, the environment, and the atmosphere in the local green spaces. These sites include three allotment sites, four community gardens, and one major local park. During the site visits, the team observed how people interact with each other and with the environment and conducted some interviews with people onsite.(2)Interviews with stakeholders (including both citizens and other stakeholders) to establish a comprehensive understanding of the schemes and pathways of GSP available in Walsall and to understand people’s experience involving GSP. The interviews with citizens were mostly delivered in small groups, and those with other stakeholders were conducted individually either via Zoom or face to face. The interview time ranges between 25 to 50 min. The interview questions concerned participants’ roles in, and experience of, GSP in Walsall and included their perceptions of the challenges and barriers to its local implementation and scaling (see Appendix A for the interview questions).(3)Observation and shadowing to generate an in-depth understanding of the daily experience of link workers and VCSEs. The researcher spent 6 h with a team of link workers (three individuals) and 9 h over 3 days with a local VCSE. During this time, the researcher observed how participants conducted consultations and delivered support to their clients. This was followed up with conversations where participants explained their experience in more detail.(4)Additionally, four workshops with a panel of key stakeholders in Walsall were conducted to support the research team to collaboratively reconstruct the framework of the local GSP system and validate preliminary findings.

As our community partners were well embedded in the community, they supported the project to recruit participants to ensure the representative of each type of ‘stakeholders’ relevant to GSP were included (as shown in Table 1). Through the research activities above, the project has engaged 34 people from three different departments from Walsall Council, two NHS partner organisations, six regional or local associations, 10 local VCSEs, four organisations hosting link workers, and two funding bodies. More than 20 have been engaged more than once.

In addition, 64 citizens were engaged through group interviews, workshops and site visits in their local parks and community centres. The demographics of the citizens shown in Table 2 are representative of the spread of citizens who are actively involved with nature and nature-based activities in the areas of the field work. The sample of citizens is not strictly comparable to the demographics of Walsall’s population; however, the sample does have sufficient representations of different genders, ages, ethnicities, and social demographics.

All research activities were conducted with informed consent being obtained. Data protocol, including anonymisation and secure storage, was strictly followed. The interviews and workshops were audio recorded or recorded on Panopto, and transcription occurred via Panopto or Otter software packages, being corrected where necessary to ensure accuracy.

## 3. Results

This study depicts the GSP system as a framework that outlines the key function areas and associated stakeholder groups in relation to GSP. As shown in Figure 1, these include:Supporting local context and conditions that are important to the delivery of GSP, including promoting, funding and supporting GSP, by providing policy guidance, funding and maintaining blue and green infrastructure;Delivering the local programmes and pathways that deliver GSP, including commissioning GSP (for example, local authority and GPs), those who link people with available services through referral (e.g., link-workers), and those who provide GSP services (for example, VCSEs);Those who benefit from GSP (for example, communities and individuals).

**Figure 1 ijerph-20-06708-f001:**
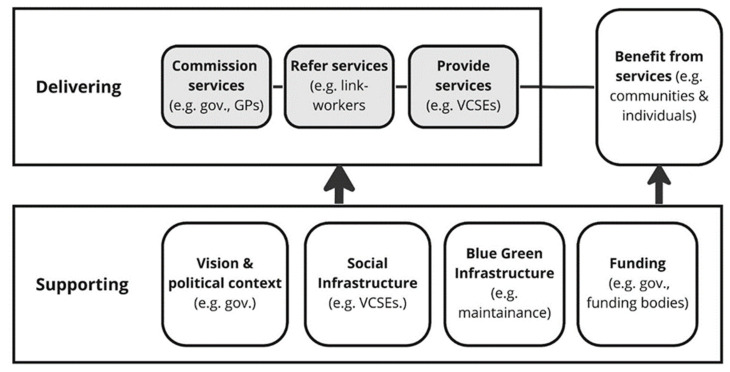
Key function areas of local GSP.

Using this framework as a guidance, the findings of the studies focus on three key aspects: (1) the supporting system, particularly the political context, the social infrastructure, blue and green infrastructure, and funding; (2) the delivering pathways; and (3) experience.

### 3.1. Supporting: Local Context

Creating a well-being borough is at the core of the local council’s agenda, as stated in the new Joint Local Health & Wellbeing Strategy (2022–2025) [88], which sets out a vision for individuals and communities to take ownership of their personal mental well-being, supported through its Integrated Care Partnerships (ICP). Walsall Together is the strategic partnership of health, social, housing, voluntary, and community organisations that are led by the Council, with the aim of supporting partners to work more closely together to tackle health inequality, focusing on not just health but the wider determinants, such as housing, education, and employment, and the vital role that people and communities play in health and well-being. Walsall has already taken some significant steps in developing support available to its residents, especially mental well-being. Introducing SP and link workers is one of these steps to help residents to access support based on their physical and mental well-being needs [91].

As local VCSEs play an instrumental role in delivering SP services, Walsall has an advantage in having a strong VCSE sector. In 2019, it was estimated that roughly 1600 registered VCSEs were operating in Walsall, a third of which focus on health and well-being as a primary service [92]. The strength of its local communities in supporting health and well-being is evidenced in how they responded to the COVID-19 pandemic. Following the “The Impact of COVID-19: Residents’ experience and wellbeing” survey [91], the Council praised Walsall’s community spirit and resilience highly. In the survey, nearly half of all respondents agreed that people in their neighbourhood pulled together to improve the local area, and a quarter of respondents volunteered or helped out in some way during the pandemic.

Nature and nature-based activities are important elements for SP. In the “Walsall Director of Public Health Annual Report: Improving Mental Wellbeing in Walsall”, accessing nature is recognised as a key measurement and an effective way to improve health and well-being [88] as it is in the ‘Walsall Green Space Strategy 2018–2022’ [93]. Nature also contributes to recovery after a crisis and to ongoing resilience. Walsall is surrounded by a green belt, and the town itself has a large proportion of green spaces in the form of parks, nature reserves, and woodland. As of 2018, there are a total of 513 different green space sites in Walsall, occupying 2178 hectares. Walsall also has many prominent and long-established community gardens, such as Caldmore Community Gardens, Goscote Greenacres Community Garden, Daffodils Community Garden, and The Butts Community Garden, and allotments, such as Borneo St, Lane Avenue, Darlaston, and Winterley Lanes. There are also many VCSEs whose focus is not specifically on green spaces or activities but which include them as part of what they offer.

In introducing SP practice, funding plays a key role in capacity building (e.g., training and hiring a network of link workers) and incentivising GPs and VCSEs to participate in SP. Long-lasting austerity measures have greatly impacted how the infrastructure, for example, green spaces, are maintained and how the services across all Council service areas are delivered. However, the level of investment has been increasing recently. In 2023, Walsall received £23.5 million from the Towns Fund. Other funding bodies, e.g., the National Lottery, also plan to provide more support and funding to this area as it is identified as an area that has historically received less investment than other regions.

### 3.2. Delivering: SP Referral Pathways

Social Prescribing (SP) in Walsall exists within an ecosystem of pathways, providers, and potential activities. There are currently three established pathways in Walsall: the Primary Care Networks (PCN), Making Connections, and Walsall Housing Group. Each pathway has its own SP team, consisting of link workers (social prescribers, or social connectors) who work with citizens to explore their needs and goals and connect them to relevant activities in the community. These pathways are summarised in Table 3 below.

#### 3.2.1. The Three Pathways

Table 3 identifies a number of similarities between the three pathways. Each has identified funding routes and measurement tools, alongside a recognised and established set of referral sources, which ranges from in-house (community housing officers and healthcare professionals) to cross-county, when referred by schools or triaged by the Midlands Fire Service, along with the option of self-referral. However, Walsall is an interesting case in that SP referrals are distributed across three main pathways both within and outside of the local PCN system. The unique setting of each pathway means that each one has its own advantages and challenges.

#### 3.2.2. Uniqueness of Each Pathway

The uniqueness of Pathway 1 (PCN) is that PCN link workers have access to the health database of people they are working with, including access to patient electronic records used by inpatient and community healthcare professionals. This means that PCN link workers are better integrated into the healthcare system and are able to track the patient’s health progress and work along with healthcare professionals to provide holistic care to a patient. For instance, for a patient who is under a PCN SP team and community mental health team, PCN link workers can track the patient’s progress with the mental health team through electronic patient notes. If the link worker detects deterioration in the patient’s mental health, they would be able to contact the patient’s community mental health support worker to see the patient and provide treatment. This integration with the healthcare system can provide holistic care for patients, detect problems early and give support in a timely manner. In comparison, the two other SP pathways are not well integrated with the healthcare system.

The second pathway (Making Connections) is funded by the Walsall Council and has its initial remit to provide services for older adults. The uniqueness of this pathway is that the Link Workers are situated within the community hubs based around the local areas. It has a good geographic coverage and receives referrals from a wider range of sources, including GPs, fire service, probation service, and community workers. Midlands Fire Service process the referrals on the DCRS database (the recording system used by the NHS) before allocating them to a hub, at which point there is an option to have a safety visit from the fire service. It has a stronger link with the local VCSEs and also outsources some services to local VCSEs on a regular basis.

The third pathway (whg) sits within the largest housing association in Walsall. One of the distinguishing features of Walsall is that it has a larger than average concentration of housing provision based within the one organisation, Walsall Housing Group (whg). It is a well-connected organisation with over 800 employees and 21,000 homes. Being a part of a large and well-connected organisation gives this pathway a unique position, allowing link workers to be directly involved with an individual’s essential needs, such as housing, that impact many of the wider determinants of health. This helps with identifying people who need help, and with being able to refer to their internal team of link workers who can provide relevant social support.

#### 3.2.3. Outside of These Pathways

It is worthwhile noting that apart from these three established SP pathways, there are other organisations operating at regional and national levels that provide SP services to Walsall; for example, Active Black Country, a funded project by Sport England aiming to tackle regional health inequality challenges through encouraging people to be physically active, has link worker capacities built within the project at each council in the Black Country region, including Walsall. Like the three Walsall-based SP pathways, it has its unique strategic context, focus, and referral routes.

### 3.3. Benefit from GSP: Experience

The majority of local people who are actively involved in nature-based activities do not have any referrals. Most had not heard of GSP, and they accessed information about activities through the internet and social media groups, local mosques, after school groups, newsletters and advertisements, friends and family, or simply word of mouth. The range of activities is diverse, including, for example, walks on the canal, picnics, after-school clubs, radio-controlled cars, kayaking with the Canals Trust, paddle boating, and high tree-top climbing. People are self-motivated to take part in these activities driven by various reasons, including personal health, and family needs. Many people we talked to were aware of the facilities and benefits: “*A third of Walsall is green spaces. We need to make the most of these assets…*” (Participant 76, male, white, 25–64) and recognize that “*you’ve got to use them or lose them… by engaging people to get into the parks, into the open spaces, into nature it benefits not only the green space but also benefits the surrounding community and area…*” (Participant 67, male, white, 25–64).

However, there are barriers, including safety and access issues. People commented that “*We have got a local park a stone throw away, but there is broken glass there, broken bottles, dog dirt*” (Participant 3, female, British Asian, 25–64), that “*I dread my children going outside because of the weapons people carry, drugs, smoking… it is scary. For kids to access nature, there needs to be… activities that are supervised*” (Participant 9, female, British Asian, 25–64)*,* and that “*I wouldn’t come to the park on my own, always with a group*” (Participant 6, female, British Asian, 25–64).

Sometimes fears are based on judgements that may prove unfounded, as is in the case where young local people had planted trees for the community orchard: “*The kids that they say will tear down all trees, have done this*” (Participant 84, male, white, 65+). And some ways of working with the community needed to be rethought: “*if you ask us we will do it, but if you tell us to do it, you’ve got a fight on your hands; don’t tell what you’re going to do to us, but let us do it.*” (Participant 84, male, white, 65+). Some issues are Walsall-wide (if not even wider): “*… it doesn’t feel very active when you walk through… Lots of unhealthy food places are able to thrive and if you want something healthy in Walsall it’s really hard to find…*” (Participant 73, female, white, 25–64).

However, the study also reveals success cases, in which GSP referrals have significantly changed people’s lives. One case is ‘C’ (anonymised), a 40-year-old male with a personality disorder and stress disorder living in social housing in Walsall, who is a victim of violent domestic abuse and at risk of becoming socially isolated. Referred through one of the hubs of the Making Connections pathway, he started his journey of recovery when he felt he was being listened to and treated like a normal human being in his first meeting with his link worker. It took a few failed attempts over the course of 6 months for him to be engaged until the appropriate support was found: a nature walk group organised by a local VCSE. C has now been with the VCSE for over a year. Being able to involve himself in a group like this has completely transformed his life: “*There was a time when I felt like smashing up the flat, but I went out for a walk, and it helped me… Sometimes you don’t need a magic pill—you just need to know the tools.*” (Participant 66, male, white, 25–64). Now he feels more like a survivor than a victim, and his relationship with nature has transformed from considering nature stressful and unsafe, to enjoying being outside and advocating for it.

This success story highlights the importance of the catalytic role of a referral and the right capability, services, and support for diverse individual needs. This case clearly demonstrates the pathway and possibility for people in need to go through GSP moving from ‘patient’ to volunteer, which changes the ledger from user to at least supporter—in some cases, even to provider. The more people experience the benefits of nature and GSP, the more impetus there is for growth: “*As you see the personal benefits you want to get everyone to enjoy this experience that you’re having… There are more uses for hydroponics than growing weed in your attic!*” (Participant 70, female, British Asian, 25–64).

However, there are many people who do not manage to get through the system for similar reasons as those initially encountered by C, and there are even more people who do not know about the scheme. The need to scale is apparent.

## 4. Discussion

### 4.1. Challenges and Opportunities

#### 4.1.1. Connecting Different Referral Pathways (Horizontal Integration)

Having multiple referral routes means that individuals have multiple access points and that SP has a wider reach/exposure to the community to support more people. However, this also adds complexity for people to navigate through the system. One interviewee (a funder/commissioner) said that “*I’m struggling to find out what is going on and I work in the sector*” (Participant 85, female, white, 25–64).

As each pathway relies on their own resources, e.g., online referral system, database, and evaluation tools, this is likely to create duplications and disconnection. The institutional boundaries of these pathways mean that information sharing across three SP pathways is not facilitated, making it hard to integrate or collaborate with each other. For example, due to compliance with General Data Protection Regulations, a link worker can refer someone to another pathway but cannot share information or data on the work that has been done. It is a systemic challenge for these pathways to share the learnings, networks, and other resources.

Each pathway follows the NHS England guidance and the standard model of SP, which involves steps from referral to the delivery of SP [94,95]. All link workers are part of the National Association of Link Workers [96], a professional body of SP link workers in the UK. The purpose is to ensure that link workers have up-to-date knowledge and training to deliver high-quality services. However, the study revealed that in adapting the standardised process of referral, each pathway has developed its own way of working to suit the needs of people they work with. For example, the link workers for the PCN pathway mostly conduct their first contact with the people via telephone; whilst at whg, all link workers ensure the first meeting is face-to-face in a comfortable setting. This significantly influences the uptake rate and experience of SP. Another example is the tools used to measure outcomes differing across the board. The PCN pathway uses Personal Well-being ONS4—four survey questions that measure personal well-being—whereas whg uses the Warwick-Edinburgh Mental Wellbeing Scale. As SP is a relatively new practice, it is important to establish channels to capture and share learning with people in other parts of the system. At the moment, these channels are not available and rely on individuals’ effort in this case. The complexity of the health system adds extra barriers for people to share and collaborate.

#### 4.1.2. Strengthening the Connection between SP Deliverers (Vertical Integration)

Although the link workers we interviewed are well connected and have very good relationships with VCSEs in Walsall, information about the services provided by local VCSEs is not readily available, and in most cases, both knowledge and networks are held by individuals. Not only do link workers have to know what activities are available or suitable (something which is subjective to each individual case, citizen, or link worker), they have to know the conditions under which the activities are offered or available. For example, many activities may not be available to citizens if it is not an acute case, they are gender- or age-restricted, or they have funding restrictions such as employment status. It is often the case that link workers have to assess and compile suitable activities and resource banks relying on individual efforts.

As VCSEs often operate on limited resources, there is a clear lack of financial viability for many VCSEs to take part in SP schemes. One interviewee shared their own experience, stating the following:


*“… we can’t pay you £15 an hour to see one of your patients…a referral assessment takes an hour… the contract just isn’t viable if we have to pay those levels of room hire… someone from NHS estates is making this decision without appreciating the context or understanding what we do…”.*
 (Participant 74, female, white, 25–64)

The majority of VCSEs are not set up with SP as a focus, and it is unlikely they have the resources and competence to tailor their services to meet the complex needs of referrals. For example, a VCSE, which has, at its heart, a desire to help people connect to nature through gardening, may not have a focus on more extensive mental health provision or safety, may not have the resources to provide this, or may not want to detract from its own central mission. This leads to a patched landscape of provision and a lack of available services for referrals. There is limited effort and investment to bring local VCSEs on board.

As Humphries [97] rightly pointed out, there were fundamental differences in entitlement, funding, and delivery between the NHS and the social care system. One example is the practice of measuring impact; one interviewee said that their NHS colleagues felt that putting money into a VCSE was “*… like throwing money into a black hole because outcomes aren’t measured in the same way…*” (Participant 70, female, British Asian, 25–64).

Another disconnection is between the link workers and the clinical professionals; one link worker commented that

*“We get a lot of GP referrals that we end up not being able to get in touch with so we get the referral forms through but we are not able to actually contact the client…”*.(Participant 73, female, white, 25–64)

The need for a better integration between primary and social care is critical in delivering effective SP. A link worker’s role is supposed to fill the gap. However, in our study, some link workers felt there was a lack of understanding of SP and their roles by clinical staff and by the wider society. One link worker we interviewed said

*“Sometimes clinicians don’t believe in, there is only a biological model, sometimes we feel unvalued because we don’t have clinical or medical background, not just doctors like practice managers, non-clinical staff, they don’t understand what we do, people have left us with disrespect”*. (Participant 75, female, British Asian, 25–64)

#### 4.1.3. Widening GSP Reach to Benefit More Communities

Walsall is an area that exhibits some extremes of deprivation, a lower-than-average life expectancy, and an ethnic and cultural diversity. These characteristics affect the awareness, perception, engagement, and uptake of GSP. Although there are three SP pathways across Walsall, GSP is perceived as something of a postcode lottery, where some have easy access and others none, depending on where they live in Walsall.

The barriers to GSP in terms of physical accessibility for those with limited money, mobility, or motivation are well established in the literature to date [29,53,98,99], and this is reflected particularly in the inequality of access for particular groups of people in Walsall. The inherited social aspects of many community-based activities mean they are or are perceived to be associated with certain cultures or demographics. This results in the disparities of access to SP for ethnic and cultural minorities based on GP referral data that shows the inconsistent recording of the wider determinants of health and variations in referral patterns on a practice-to-practice basis [100]. The literature suggests that most community gardens are racially segregated, and the majority of gardeners also appear to be middle-class [101]. This is definitely the case for Walsall; some people we talked to mentioned that they had previously been made to feel unwelcome because of their ethnic or social background, saying

*“I had a personal experience at a local allotment where I didn’t feel entirely welcome… And when I started to speak to people, especially around our community centre in this area, people felt that allotments were not for them, that… there are predominantly white middle class, things that they just can’t engage with. And if they don’t know what they’re doing is going to be frowned upon”*. (Participant 70, female, British Asian, 25–64)

The social and cultural differences revealed as motivation, personal preference, and interest must be catered to. This highlights the importance of inclusive and collaborative approaches [102] as an effective way to tailor the interventions to the different needs and aspirations of people, to identify meaningful outcomes and feasible and robust methods for the uptake of SP. In the meantime, it is challenging for the system and services to be tailored to match this, highlighting the importance of joined up, adequately funded, suitable activities, and of the importance of people, their passion, knowledge, and experience, whether they are link workers or VCSEs.

#### 4.1.4. Reviewing Approaches to Enable People to Benefit from What Nature, as a Valuable Local Asset, Can Offer for Health

Although nature is an important part of local assets, which potentially creates health benefits as part of SP offerings, the priority of offering nature-based activities is perceived as low by link workers amongst many other ‘solutions’ they need urgently address, e.g., housing. Because of this, the drivers of poverty and housing will take precedence over green activities, and health inequality will persist as those in areas of deprivation are not given the opportunity to benefit from nature because of a raft of systemic pressures. LWs and VCSEs cannot solve these larger issues despite their knowledge and passion. This highlights the importance of having a more comprehensive approach to address wider economic and social determinants of health while considering GSP as one route to support the overall goal.

Another more fundamental barrier for people to access GSP is related to infrastructure. Despite Walsall being a green borough, the amount of green space in Walsall does not translate into health benefits equally across the borough. For many areas with a high population density, nearby high-quality green space is lacking. This leads to other access barriers, e.g., transportation, safety, money, efforts, and time. The inequality of access to green spaces mirrors the health inequalities present in the area. This leads to the question of whether the investment should focus on “bringing nature to people”—developing green areas in people’s nearby environment—or rather to “bring people to nature”—encouraging and facilitating people to actively participate in nature-based activities, as discussed in the literature [11,103].

#### 4.1.5. Limitations

We recognise that these findings were limited to Walsall, as the result of a place-based case study approach. The case study approach allowed us to develop a deep understanding of the phenomena in this particular geographical context and contributed to the critical review of place-based approaches in health and care. However, it does not allow the authors to assume what is shared and what is unique to this context. Therefore, we suggest future studies to focus on triangulating the results from this case study with others that are happening concurrently through other projects. Further, the demographic information of the participants, apart from ethnicity, age, and gender, were not considered as being of material importance in developing the case study. Therefore, the project did not collect this type of information (e.g., income, marital status, and education) to ensure the discussion was centred on their experience of GSP. However, we recognise that it is an important area of research to understand how the participants’ backgrounds determine their involvement and experience with nature and GSP, which could be achieved in future studies.

## 5. Conclusions

Social Prescribing (SP) was introduced to Walsall during the COVID-19 Pandemic under the context of the health system reforms, moving from a centralised system to a more community-based and personalised approach. Bringing different local assets, including nature, into health and care provision is also high in local governments’ agenda. There is a high expectation for SP to support the government’s health strategy and the recent levelling-up agenda.

The case study has revealed the local contexts, the SP pathways, and people’s lived experience. How these pathways are structured, resourced, and delivered is largely shaped by the local contexts. The findings are important in informing the future development of SP schemes. At the moment, there is no literature specifically investigating the delivery of SP in great detail. It opens up an important research question as to how the value of nature and other forms of local assets to health and health inequality can be scaled beyond SP to include wider communities. The value of referral is to bring people to community-based services they do not normally have access to. It is extremely valuable when people who need the services are identified and referred so that they can then fully engage the services in the long term, as evidenced by the success cases. The ultimate aim of providing a referral is to increase the ability of citizens to access health support. However, the benefit of referral will only be achieved when all the conditions—enjoyment, ability level, applicability, and eligibility, for example—are met. Therefore, to scale the practice to benefit a larger population requires system changes and consistent investment. This highlights an area of research to understand how various forms of local assets, e.g., culture, arts, and heritage, are included in SP in addressing the wider determinants of health and what the different challenges are associated with each.

The discussion about the challenges and opportunities reveals a need for a range of actions. First is the need for collaboration and integration between different pathways and between the social sector and the health sector. This is evidenced by the lack of a shared database and resource banks between different referral pathways and between different actors, and by the lack of mutual respect, recognition and engagement with actors outside of the NHS. Secondly, it is important to bring VCSEs on board by providing an appropriate level of investment into developing capacities and resources, as they deliver the majority of SP. Thirdly, it is crucial to act on the challenge of access inequality. Walsall is a good example to highlight this need. To enable SP to work for everyone in need, the activities, referral processes, and supporting mechanisms need to put people’s needs in the centre of the thinking. A more collaborative and genuine place-based approach is essential. Lastly, alongside GSP, investment into infrastructure is needed to move the health paradigm from “prevention” to “promotion”, so that more people can benefit from what nature can offer.

## Figures and Tables

**Table 1 ijerph-20-06708-t001:** Engagement with other stakeholders.

Local Council	
Resilient Communities, Walsall Council	Director
Public Health Development, Walsall Council	Senior Manager
Healthy Spaces, Walsall Council	Team Lead
NHS partners	
NHS Black Country	Innovation and Development Manager
Walsall Together	Programme Lead
Walsall Together	Transformation Programme Manager
Regional and local organisations	
Walsall For All	Head of Community, Equality and Cohesion in Walsall
One You Walsall	Service Manager
One You Walsall	Programme Manager
One Walsall	Operations Manager
Brownhills Community Association	Centre Manager, Head of Community Network,
Active Black Country	Head of Insight: Health and Well-being
British Triathlon Association	Regional Programme Manager
Local VCSEs	
Goscote Greenacres	Community Garden Manager
The MindKind Projects	Founder/CEO
Caldmore Community Gardens	Community Garden Manager, Volunteers, Citizens × 2/3 visits
Winterley Lane Allotments	Secretary, Citizens
Lane Avenue Allotments	Citizen/participant of Lane Avenue Allotments Secretary
Borneo Street Allotments	Secretary
Butts Community Garden	Secretary
Daffodils Community Garden	Secretary
Darlastan All Active Garden	Manager
Birchills Agenda 21	CEO
Link workers	
PCN Walsall South 2	Senior Social Prescriber/Link Worker
Walsall Housing Group (whg)	Link Worker
Making Connections Walsall, Bloxwich Community Partnership	Social Connector/Link Worker
Health Exchange	Community Connector
Funding bodies	
National Lottery Heritage Fund	Engagement Manager
Bumblebee Conservation Trust	Outreach Officers

**Table 2 ijerph-20-06708-t002:** Engagement with citizens.

Ethnicity	Percentage
Asian, Asian British, or Asian Welsh	33%
Black, Black British, Black Welsh, Caribbean, or African	2%
Mixed or Multiple ethnic groups	2%
White	61%
Other ethnic groups	3%
Genders	
Male	25%
Female	75%
Age Bands	
17–24	20%
25–64	50%
65+	30%

**Table 3 ijerph-20-06708-t003:** Three established social prescribing pathways in Walsall.

	Pathway 1: Primary Care Network (PCN)	Pathway 2: Making Connections	Pathway 3: Walsall Housing Associates (whg)
Funder	Additional Roles Reimbursement Scheme (ARRS) of PCN	Walsall Council	whg and external funding, e.g., National Lottery
Measurement	Personal Well-being ONS4	ONS4/Warwick-Edinburgh Mental Wellbeing Scale (WEMWBS)	Warwick-Edinburgh Mental Wellbeing Scale (WEMWBS)
Starting year	2019	2017	2020
Strategic context	Part of the nation-wide SP development within PCN	Funded projects with the initial aim of reducing loneliness and isolation in people of 60+ years	Part of whg’s corporate strategy plan for 2020–2024 to deliver ”H Factor programme” (Health, Hope, and Happiness)
Capacity	14 link workers distributed amongst 7 PCNs dependent on need	Operating via four community hubs placed around Walsall. Each hub has 1–2 link workers working different shift patterns	6 link workers and a team manager
Source of referrals	Primary care professionals, GPs, practice staff, community pharmacists, and self-referrals via PCN website	Primary care and other healthcare professionals, e.g., GPs, fire service, probation service, community workers, and self-referrals via Walsall Council’s website	whg internal teams, e.g., community housing officers; healthcare professionals and self-referrals via whg’s website
Advantages	- have access to the health database - are better integrated into the healthcare system	- situated within the community hubs based around the local areas with some services outsourced to other VCSEs and groups.	- being a part of a large organisation that is well connected and directly involved with individuals’ essential needs to be able to address many of the wider determinants of health

## Data Availability

Due to the ethically and politically sensitive nature of the research, participants of this study did not agree for their data to be shared publicly. Please contact the project PI at qian.sun@rca.ac.uk to discuss the underlying data.

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
