# Peer review of "Green Social Prescribing in Practice: A Case Study of Walsall, UK"

_ijerph, 2023, doi:10.3390/ijerph20176708_

Round 1
Reviewer 1 Report
Green Social Prescribing in Practice: A case study of Walsall, UK
Thank you for allowing me to serve as a reviewer on this very interesting paper which presents a case study of Green Social Prescribing (GSP) in Walsall, UK. By introducing this case study the paper addresses specific gap of knowledge and explores how GSP is implemented.
Given that we know little about the mechanisms that actually affect the implementation
Of GSP in the local government level, the current research is meaningful with the potential to make theoretical and practical contributions for both public health and environmental literature. However, it has some important issues that should be modified. I did have some concerns with regard to the methodology section. In step, I have provided a few comments to, hopefully, help with further revisions of the article.
General:
1. The strength of this paper lies in the specific focus on GSP in practice in Walsall.
Precisely, it captures the initial learnings from the local initiative and explores how the challenges, barriers, and opportunities identified in the literature are played out at the local level.
2. The weakness, however is that the research question is not clearly presented either in the introduction or in the background. Also, perhaps further specify how the qualitative research design contributes adequately to the understanding of the phenomenon.
Introduction:
1. Well-written and well-structured section.
2. One weak point in this section is that it is not cover previous works regarding the GSP at the local level and how the current article corresponds with previous studies or reinforces their findings.
Methodology:
I find the methodology to be a good attempt, but the weakest aspect of the paper. Here some suggestions to improve this section:
1. Case study approach – please elaborate the rational for this method.
2. This whole part must be more focused. Different kinds of methods used to investigate this case, but without proper synchronization in the findings section.
3. Please stress why interview data are most suited to understand the local context.
4. Please provide some sample questions asked to the participants (add the instrument: interview guideline)
5. Please elaborate specifically which steps you have undertaken to guarantee rigor, experiences, skill and relevance in terms of reliability and validity.
6. Please mention other aspects of ethical conduct, such as informed consents (for the interviews) and data+data storage.
Results:
1. Findings must be presented for each kind or research method used in the study.
Maybe using a table?
2. What are the themes that emerged from the analysis of the interviews (as mentioned – "In total, 62 interviews with citizens and 30 with other stakeholders 233 were conducted. The average interview time was 15 minutes with citizens and 50 minutes with other stakeholders").
Conclusions:
1. The conclusions are brief and to the point, which is a strong feature.
2. Missing - a paragraph detailing the limitations of the study
Author Response
Thank you for allowing me to serve as a reviewer on this very interesting paper which presents a case study of Green Social Prescribing (GSP) in Walsall, UK. By introducing this case study the paper addresses specific gap of knowledge and explores how GSP is implemented.
Given that we know little about the mechanisms that actually affect the implementation of GSP in the local government level, the current research is meaningful with the potential to make theoretical and practical contributions for both public health and environmental literature. However, it has some important issues that should be modified. I did have some concerns with regard to the methodology section. In step, I have provided a few comments to, hopefully, help with further revisions of the article.
General:
- The strength of this paper lies in the specific focus on GSP in practice in Walsall. Precisely, it captures the initial learnings from the local initiative and explores how the challenges, barriers, and opportunities identified in the literature are played out at the local level.
Thank you for clearly noting down the contribution of the paper.
- The weakness, however, is that the research question is not clearly presented either in the introduction or in the background. Also, perhaps further specify how the qualitative research design contributes adequately to the understanding of the phenomenon.
Lines 48-50 now introduce the qualitative research design, and the methodology section is extended to explain the research design.
Lines 68-69 now present the research question.
Introduction:
- Well-written and well-structured section.
- One weak point in this section is that it is not cover previous works regarding the GSP at the local level and how the current article corresponds with previous studies or reinforces their findings.
We’ve added extra texts (lines 71-74) to explain more explicitly in the Introduction how this paper adds to previous research that further strengthens the originality summarised as the gap of knowledge (lines 628-631) identified through literature review. The new extended methodology (lines 210-245) further explains how the case study adds to the knowledge.
Methodology:
I find the methodology to be a good attempt, but the weakest aspect of the paper. Here some suggestions to improve this section:
- Case study approach – please elaborate the rational for this method. (specify how the qualitative research design contributes adequately to the understanding of the phenomenon)
Section 2.1 (lines 210-245) has now been much extended to address the comments re methodology. In this section, we have explained why case study is chosen to answer the research question, and how the interviews and other methods were used to collect data to guarantee reliability and validity. We have covered ethical considerations and added sample questions in new tables.
- This whole part must be more focused. Different kinds of methods used to investigate this case, but without proper synchronization in the findings section.
We’ve added texts (line 222-237) to explain how the data analysis has been conducted. As explained, a panel of multidisciplinary researchers and practitioners (authors) were established to analyse the data. This allows us to identify new knowledge and practice. The process of triangulation allows us to develop the conclusions in a very rigorous way.
- Please stress why interview data are most suited to understand the local context.
Please see the answer for Q1.
- Please provide some sample questions asked to the participants (add the instrument: interview guideline)
Interview questions (Appendix 1) added.
- Please elaborate specifically which steps you have undertaken to guarantee rigor, experiences, skill and relevance in terms of reliability and validity.
Please see the answer for Q1.
- Please mention other aspects of ethical conduct, such as informed consents (for the interviews) and data+data storage.
Please see the answer for Q1.
Results:
- Findings must be presented for each kind or research method used in the study. Maybe using a table?
In the Methodology section, we have added details to explain how case study as a method was used and the data has been analysed to develop the case study. As explained in the new methodology section (lines 222-230), the case study approach was used to yield in-depth place-based information where thematic analysis is less relevant. Instead, we triangulated the findings according to the research objectives.
- What are the themes that emerged from the analysis of the interviews (as mentioned – "In total, 62 interviews with citizens and 30 with other stakeholders 233 were conducted. The average interview time was 15 minutes with citizens and 50 minutes with other stakeholders").
See above.
Conclusions:
- The conclusions are brief and to the point, which is a strong feature.
- Missing - a paragraph detailing the limitations of the study
We’ve added a new section 4.1.5 Limitations (lines 611-617)
Reviewer 2 Report
I appreciate Green Social Prescribing manuscript. I think, we need more descrioption as follows; 1) what is green social prescribing in terms of operational definition 2) who funded green social prescribing project 3) Could you pls describe about green social prescribing activity places?
Author Response
I appreciate Green Social Prescribing manuscript. I think, we need more description as follows;
- what is green social prescribing in terms of operational definition
The definition of green social prescribing is from lines 28-33.
- who funded green social prescribing project
Social prescribing projects are generally funded by the NHS as part of the health services provided and delivered by the local VCSEs and NHS’s health partners. In the Introduction in defining Green Social Prescribing (lines 28-33), this has been clearly stated.
However, if the reviewer is asking who funded this research project, it was funded by Arts and Humanities Research Council (AHRC) as explained in lines 272-275.
- Could you pls describe about green social prescribing activity places?
When explaining the concept of Green Social Prescribing, examples were given to describe the type of activities and where these activities normally take place in lines 33-36.
Reviewer 3 Report
This is a very well-written paper, but there are some areas where the paper requires clarification and also the voice of the people who make use of green spaces and/or GSP in Walsall needs to be more visible in the manuscript. This would considerably strengthen the paper. Otherwise it seems such a shame that such an extensive set of interviews with participants are not really being used in this manuscript.
1. In the introduction please clarify for international readers whether there are any out of pocket costs for people referred to GSP, or whether the costs of their participation are fully paid by their GPs, or entirely to free to everyone, e.g. park runs. This is not something is discussed later in the paper where the focus on funding seems to be on funding for providers rather than participants.
2. Line 83: Need to revise to make clear that the NHS Strategy is for England only and not the UK.
3. Lines 87 to 89 – the authors talk about the roll out of social prescribing across the UK, can they provide evidence that it is being rolled out in all four countries of the UK, each with their own devolved responsibility for health, or otherwise make clear that the roll out is just in England?
4. Line 104/105 – so the positive evidence does not apply to people of working age?
5. The authors need to make clear when the study was undertaken. They note it lasted 12 months, but do not provide the time period of the study, including making it clear whether this took place after all COVID restrictions had been lifted?
6. Clarify the means of interview I think most were face to face but unclear if all were
7. More information on members of public interviewed, what is the breakdown gender, age and possibly ethnicity as this seems relevant here. Clarify if interviews recorded and transcribed or if note how were interviews documented? Were they all individual interviews or were there some group interviews?
8. Very importantly a key weakness in the paper as it currently stands is that 62 members of the local community that might used GSP were interviewed – but their views are not well described in the paper. I would have like to have seen quotes from these people (indicating gender and age (and possibly ethnicity)) in the paper in relevant places to support the conclusions drawn. This seem somewhat lacking. What do local people think about the appropriateness of these schemes and how they work? This is most obvious in the section 3.3 – what did people actually positively and negatively about their experiences both of SP and of Nature?
9. Another place where quotes would also help is the section 4.1.3 where it is stated that “some people we talked to mentioned that they had previously been made to feel unwelcome because of their ethnic or social back-ground.” So what did they actually say?
Author Response
This is a very well-written paper, but there are some areas where the paper requires clarification and also the voice of the people who make use of green spaces and/or GSP in Walsall needs to be more visible in the manuscript. This would considerably strengthen the paper. Otherwise it seems such a shame that such an extensive set of interviews with participants are not really being used in this manuscript.
- In the introduction please clarify for international readers whether there are any out of pocket costs for people referred to GSP, or whether the costs of their participation are fully paid by their GPs, or entirely to free to everyone, e.g. park runs. This is not something is discussed later in the paper where the focus on funding seems to be on funding for providers rather than participants.
This has been further clarified in lines 51-55
- Line 83: Need to revise to make clear that the NHS Strategy is for England only and not the UK.
We have changed ‘UK’ to ‘England’ in relevant places where the UK was referred to. Line 83 is now line 99.
- Lines 87 to 89 – the authors talk about the roll out of social prescribing across the UK, can they provide evidence that it is being rolled out in all four countries of the UK, each with their own devolved responsibility for health, or otherwise make clear that the roll out is just in England?
We have clarified throughout the Introduction that Social Prescribing is being rolled out in England (line 58). This is also considered as the limitation of the paper that suggests future studies to look into whether and how social prescribing has been rolled out across the UK and probably many other countries.
- Line 104/105 – so the positive evidence does not apply to people of working age?
Changes made to clarify that this applies to people in general (now lines 117-118).
- The authors need to make clear when the study was undertaken. They note it lasted 12 months, but do not provide the time period of the study, including making it clear whether this took place after all COVID restrictions had been lifted?
Time period of the project has been added in lines 275-278.
- Clarify the means of interview I think most were face to face but unclear if all were
In line 290, further clarification was made to explain the interviews were both face to face and online, depending on the interviewees’ preference.
- More information on members of public interviewed, what is the breakdown gender, age and possibly ethnicity as this seems relevant here. Clarify if interviews recorded and transcribed or if note how were interviews documented? Were they all individual interviews or were there some group interviews?
We have clarified the sampling and interview processes in lines 290-297.
- Very importantly a key weakness in the paper as it currently stands is that 62 members of the local community that might used GSP were interviewed – but their views are not well described in the paper. I would have like to have seen quotes from these people (indicating gender and age (and possibly ethnicity)) in the paper in relevant places to support the conclusions drawn. This seem somewhat lacking. What do local people think about the appropriateness of these schemes and how they work? This is most obvious in the section 3.3 – what did people actually positively and negatively about their experiences both of SP and of Nature?
Quotations of the interviews were added throughout Section 3.3: please see highlighted passages.
- Another place where quotes would also help is the section 4.1.3 where it is stated that “some people we talked to mentioned that they had previously been made to feel unwelcome because of their ethnic or social back-ground.” So what did they actually say?
Added in Section 4.1.3, lines 575-579
Round 2
Reviewer 1 Report
With the first version I was sceptical, but I am glad to see that the authors have submited an improved revision of the manuscript.
To accept in present form.
Author Response
Thank you.
Reviewer 3 Report
The authors have responded reasonably well to my comments. However, there is still no information provided on the age and gender breakdown or participants as well as informaiton on ethnicity. The age and gender breakdown do need to be provided; if there were not collected by the authors this is an important limitation that needs to be highlighted in the limitations section, as it means the results are not transparent.
I thank the authors for adding some quotes, however they still do not indicate the age and gender of the respondent, or provide a unique identifier so as we know how many unique individuals that these quotes come from. This is needed. Again without this the results are not sufficiently transparent. Each quote should be clearly accompanied by the age, gender and unique but anonymised identifier of the respondent.
Author Response
We welcome the opportunity to respond to the reviewer’s concern about the process of data collection and analysis. We appreciate that this is an important factor determining the significance of the findings and have thoroughly reviewed the data again and made the following changes to the paper:
- Add table 2 to show the breakdown of citizens engaged in the project
The project used a number of methods to collect data as explained in the methodology section, including workshops, observations / site visits, and interviews to develop the case study.
For citizens engaged in the project, their basic demographic information (age band, gender, and ethnicity) was collected.
For participants representing other stakeholders, information about their roles and organisations was collected on top of the demographic information, which allows us to ensure the sufficient reach of different stakeholders relevant to GSP as shown in Table 1.
Each participant has been given a unique identifier following the usual research practice.
We didn’t include the demographic information of citizens in the original paper, considering this type of information as less relevant to the discussion. However, we have added a summary of the demographic information in Table 2. We hope this helps to assure the reviewer that the project has followed a rigorous research process. We are in the process of depositing the transcripts of interviews and workshops to AHRC’s recommended site.
- Add participant identifiers and demo information following each quotation
We have added participant’s identifier and demographic information for each quotation cited in the paper as suggested by the reviewer.
- Rewrite methodology to explain different methods used to collect data and sampling process
We have also rewritten parts of the methodology to clarify that the project has used different types of data collection methods (that are not limited to interviews of citizens) and thus the data collected may be presented in different formats, but that the richness of data allows the researchers to develop in-depth understanding of the place, the practice, and people’s experience, as a case study.
- Add lines in limitation
We have also reflected on the possibility of collecting more demographic information about the citizens and suggest future studies to focus on this.
We hope this explains how we have considered the comments and hope these changes address the concerns the reviewer has.